# The influence of age, gender and pharmacogenetic profiles on the perspective on medicines in the German EMPAR study

Veronica Atemnkeng Ntam[1]©, Tatjana Huebner[1]©*, Michael Steffens[1], Christoph Roethlein[1], Britta Haenisch[1,2,3], Julia Stingl[3,4], Roland Linder[5], Catharina Scholl[1]

1 Research Division, Federal Institute for Drugs and Medical Devices, Bonn, North Rhine-Westphalia, Germany, 2 German Center for Neurodegenerative Diseases (DZNE), Bonn, North Rhine-Westphalia, Germany, 3 Center for Translational Medicine, Medical Faculty, University of Bonn, Bonn, North Rhine-Westphalia, Germany, 4 Institute for Clinical Pharmacology, RWTH Aachen University, Aachen, North Rhine-Westphalia, Germany, 5 Techniker Krankenkasse (TK), Hamburg, Germany

© These authors contributed equally to this work.
* huebner.ta@gmail.com

**Data Availability Statement:** All relevant data are within the paper and its Supporting Information files.

## Abstract

### Background

Pharmacogenetic testing in routine care could provide benefits for patients, doctors and statutory health insurances. Therefore, the aim of the retrospective, observational study Einfluss metabolischer Profile auf die Arzneimitteltherapiesicherheit in der Routineversorgung (EMPAR) was to analyze the relationship between pharmacogenetic profiles, the risk of adverse drug reactions, and patients' perceptions of drug therapy in 10748 adult ($\geq$18 years) participants in Germany.

### Methods

A questionnaire was used to assess views and beliefs about medicines and participants individual perception of sensitivity to drug therapies. The questionnaire consisted of the Beliefs about Medicines Questionnaire (BMQ)-General scales (Overuse, Harm, Benefit), the Perceived Sensitivity to Medicines (PSM), Natural Remedy, and Gene Testing scales. The influence of gender, age, study collective, genotype and phenotype of relevant pharmacogenes on participant's perception were evaluated.

### Results

Overuse, PSM and Benefit scores were significantly higher among patients of the collective International Classification of Diseases and Health Related Disorders (ICD)-10 Y57.9! diagnosis, which indicates complications related to drugs, compared to the anticoagulant/antiplatelet and cholesterol-lowering drug collective. Age and gender also played a significant role in patients' perceptions, with younger patients and female participants more likely to believe in medication overuse according to the Overuse scale score compared to older and male participants. Female participants compared to male participants and the old age group

**Funding:** This EMPAR study was funded by the Innovation Fund of the Federal Joint Committee in Germany (Gemeinsamer Bundesausschuss). Grant number: 01VSF16047. The funders had no role in study design, data collection and analysis, decision to publish, or preparation of the manuscript.

**Competing interests:** The authors declare no conflict of commercial or financial interests.

compared to the young and/or middle-age subgroup, scored higher in PSM and/or Harm scales, respectively. Only a tendency of increased Harm, Overuse and PSM scores was observed in the participant group with five or more relevant actionable variants compared to subgroups with 0 up to 4 variants.

## Conclusions

In conclusion, patients' beliefs about medicines and their drug sensitivity perceptions are influenced by various factors including age, gender, previous complications with medicines, and with some tendency also pharmacogenetic profiles. The higher association with more negative views related to treatment indicates that there is a need to target the underlying issues in affected patient groups in order to improve compliance to treatment and outcomes in routine care.

**Trial registration:** EMPAR was registered in the German Clinical Trials Register (DRKS) on 06 July 2018 (DRKS00013909).

## Introduction

The German retrospective, observational study EMPAR investigated the potential effectiveness of pharmacogenetic testing in routine clinical care analyzing pharmacogenetic profiles, health care claims data and the patients' attitude towards medicines on the basis of the Beliefs about Medicines Questionnaire (General) by Horne et al 1999, towards natural remedies, pharmacogenetic testing and their sensitivity to medicines [1, 2].

Identifying possible risk factors prior to drug therapy can help to minimize the likelihood that drugs will cause adverse drug reactions (ADRs). Among medical errors and drug interactions that may cause ADRs, a patient's pharmacogenetic variability plays an important role in the success of the drug therapy [3]. Patients may have an ultra-rapid, extensive, intermediate or poor metabolism in terms of one or even a combination of important pharmacogenes for a drug therapy. This can have a significant impact on the efficacy and safety of a treatment [4]. Another major factor that contributes to the success of drug therapy is the patient's individual beliefs about medicines [4, 5]. It appears to strongly correlate with the patient's consistency concerning compliance to the pharmaceutical therapy in question. This consistency can be greatly influenced by the individual experience and knowledge [6].

Horne et al. established the BMQ in 1999 as a tool to measure the patients' views and values on medicinal products. BMQ-General is available in two versions. The BMQ-G8 version is subdivided into the scales Overuse and Harm. Each has four items, which assess a patient's view on how often doctors prescribe medicines or to what degree medicines are viewed as harmful by patients respectively. The second BMQ version, BMQ-G12, has four additional questions (General Benefit), which are applied to assess a more positive view on the benefits of pharmaceuticals [7, 8]. So far, the BMQ is available in several languages [9–11].

The German version of the BMQ-General was e.g. used to assess the drug therapy adherence in a German primary care setting among patients with a series of chronic illnesses [12]. For the EMPAR study, BMQ-G12 version was used. In addition, the Perceived Sensitivity to Medicines Score (PSM with four questions), was included as the fourth block of the questionnaire. It assesses a patient's view on the personal sensitivity towards medicine [13]. A Patient's sensitivity perception towards medicines might influence the kind of treatment they choose to

follow and adhere too. A strong sensitivity perception might likely cause a patient to alter treatment dosages and or stop treatment completely due to fear of an adverse event [14, 15].

It is estimated that about 5.3% of all hospital admissions in Germany are potentially related to adverse drug reactions (ADRs) [16, 17]. Furthermore, people are increasingly turning to herbal or "natural" remedies because they believe plant remedies are free from undesirable side effects [18, 19]. However, this assumption, often leading to unmonitored self-medication, is not supported by current evaluations and therefore is misleading as safety is not warranted because the source of the remedy is "natural" [18]. The Natural Remedy Scale (three questions) was included in the present evaluation to assess participants' beliefs about natural remedies.

A further one question item, "Gene Testing", was also included in the EMPAR questionnaire. Precision medicine has made extensive use of gene testing. It provides an important application for pharmacogenomics guided pharmacotherapy, which adjusts drug choice and dosage based on a patient's genetic characteristics. International scientific consortia working to develop medical guidance for the clinical application of pharmacogenomics have been able to develop treatment guidelines as a result of the recognized role that pharmacogenomic variation plays in therapeutic efficacy and safety to date [20–22]. Through the identification of genetic variants that can elevate the likelihood of adverse drug reactions, healthcare professionals can enhance their ability to make well-informed choices regarding both the selection and dosage of medications [23, 24]. However, the willingness of patients to undergo a pharmacogenetic examination in routine care is crucial for the success of such therapeutic guidance in Germany and needs to be investigated. The willingness to participate in gene testing for therapy management therefore was addressed in the present evaluation.

Eligible patients from the EMPAR study took part in the survey [1, 2]. The evaluated population consisted of three genotyped collectives; patients with an initial prescription of antiplatelet or anticoagulant drugs, cholesterol-lowering drugs and patients with at least one ICD -10 Y57.9! diagnosis that indicates complications that have occurred due to medicines or drugs.

The aim of the present evaluation was to analyse patients' general views on medicines and natural remedies, their willingness to participate in pharmacogenetic testing in clinical settings and their perception of sensitivity to drug therapies. Furthermore, a major issue was to evaluate whether such attitudes are affected by metabolic or pharmacogenetic profiles respectively. Here, we present the outcomes of the identified association of age, gender and pharmacogenetic profiles with the evaluated EMPAR participant's beliefs.

## Methods

### Ethics

The study protocol was approved by the Ethics Committee of the Medical Faculty at the University of Bonn. All study participants provided written informed consent.

### Analysed study population

The recruitment procedure of eligible adult policy holders of the statutory health insurance Techniker Krankenkasse (TK) for the EMPAR study was published by J. Fracowiak et al. in 2022. Recruitment started on 24.07.2018 and ended on 30.04.2020. Last exclusions due to missing data or insufficient quality were performed until 31.12.2020. All enrolled policy holders provided written informed consent to participate in the study. Participant data of one year prior to initial prescription of the drugs of interest or the Y57.9! diagnosis, the year of an initial prescription or the Y57.9! diagnosis if it was detected in 2013–2018 and data of one to two subsequent years was provided by the involved health insurance company. All evaluated data sets were anonymous. Authors had no access to information for identification of individual

participants during or after the collection of data. Quality assessments started during recruitment. Assessments of the complete study population started in January 2021. In order to determine whether pharmacogenetic testing may benefit patients in Germany, the EMPAR (Influence of Metabolic Profiles on the Safety of Drug Therapy in Routine Care/ German: Einfluss metabolischer Profile auf die Arzneimitteltherapiesicherheit in der Routineversorgung) project examined whether it could potentially predict the risk of ADRs or therapy resistances and lower healthcare costs. Furthermore, a questionnaire mainly on the basis of the BMQ, was included in the assessments to evaluate participants' general views and beliefs about medicines. EMPAR participants who submitted an answered questionnaire were matched on a 1:1 basis with enrolled EMPAR individuals who also provided a pharmacogenetic profile of sufficient quality. Participants whose pharmacogenetic data did not meet up with quality control metrics or who withdrew participation were excluded from the assessments. Furthermore, for this evaluation participants who only partially answered or did not answer questionnaire block items were excluded. The evaluated questionnaire population further consisted of three collectives; participants with an initial prescription of anticoagulants or antiplatelets, cholesterol-lowering drugs and patients with at least one ICD-10 Y57.9! diagnosis.

## EMPAR questionnaire

For the EMPAR study, the BMQ-G12 version (Overuse scale: 4 questions, Harm scale: 4 questions, General Benefit scale: 4 questions) was used together with the Perceived Sensitivity to Medicines scale (PSM scale: 4 questions) in German [13, 25]. Two additional question blocks in German were adapted from the evaluation by Jörg Sonnabend from the University of Ulm research team [26] and included into the EMPAR questionnaire: The Natural Remedy scale (three questions: "Herbal medicines are more tolerable than chemically produced ones", "Herbal medicines work better than chemically produced ones", "If possible, I would prefer a herbal medicine") and Gene Testing (one question: "I would have my genes tested if it meant I could know which drug would suit me best").

The answers are scored using a five-point Likert scale ranging from 1 = (do not agree at all) to 5 = (fully agree). The BMQ-General 12 and PSM scoring can either be lowest 4 points to highest 20 points, meanwhile the Natural Remedy scale can either be scored lowest 3 points and highest 15 points and in the case of Gene Testing, the lowest score is 1 and the highest is 5. Higher scores are indicative of a stronger and lower scores indicated a milder belief.

## Questionnaire population and stratified scale score analyses

**Consistency, descriptive and statistical analysis.** Cronbach's Alpha ($\alpha$) was measured to evaluate the internal consistency and correlation among the applied scales. Cronbach's alpha quantifies the level of agreement on a standardized 0 to 1 scale. According to George and Mallery 2019, values of $0.9 > \alpha \geq 0.8$, $0.8 > \alpha \geq 0.7$ and $0.7 > \alpha \geq 0.6$ are "good", "acceptable" and "questionable", while values of $0.6 > \alpha \geq 0.5$ and $0.5 > \alpha$ can be interpreted as "poor" and "unacceptable" respectively [27]. The Pearson correlation coefficient based on the evaluated scale scores was assessed as follows: Harm vs Overuse, Harm vs PSM, Overuse vs PSM and finally Harm, Overuse and PSM each vs Benefit.

The mean, standard deviation (SD), median and interquartile range was calculated for Harm, Overuse, Benefit, PSM as well as Natural Remedy scale scores. The Gene Testing scale consisted of only one question. Therefore, the Gene Testing score was analysed, however not in comparison to the BMQ-G12 or PSM scale scores. For the evaluation of the complete EMPAR questionnaire population and a comparison of questionnaire study cohorts in terms

of beliefs about medicines, sensitivity perception, natural remedies and gene testing, a stratification by age or gender was considered.

Statistical analyses were performed in IBM SPSS Statistics for WINDOWs, version 28.0.

**Genotype and phenotype evaluations.** Genotypes and phenotypes of relevant pharmacogenes in the EMPAR population were evaluated in detail by Fracowiak et al. in 2022 (2). For scale score evaluations of patients stratified by the number of actionable variants (see Fracowiak et al. (2022)), one-way ANOVA analysis was performed to compare whether differences between group means were significantly greater than within group differences. Participants were evaluated as carriers of actionable variants if they were homozygous or heterozygous in terms of mutant alleles of the SNPs of interest (*Cytochrom P450 (CYP)2C19*: rs12248560, rs41291556, rs4244285, rs4986893; *CYP2C9*: rs1057910, rs1799853, rs28371685, rs28371686, rs56165452, rs7900194, rs9332131; *Solute Carrier Organic Anion Transporter Family Member 1B1 (SLCO1B1)*: rs4149056; *Vitamin K epOxide Reductase Complex subunit 1 (VKORC1)*: rs9923231. A two-tailed t-test or one-way ANOVA and Tukey post-hoc Test was performed to determine whether the means of the scale scores were significantly different between three age groups (young adults of <36 years, middle age participants of 36–60 years and old participants of >60 years), between male and female questionnaire participants, in participants with extreme phenotypes compared to normal phenotypes of *CYP2C9* and *CYP2C19*, and furthermore between questionnaire collectives.

# Results

## Enrolment of participants

A total of 10899 insurees of the statutory health insurance Techniker Krankenkasse (TK) participated in the EMPAR questionnaire survey with 8455 (77.6%), 1922 (17.6%) and 522 (4.8%) in the anticoagulant/antiplatelet, cholesterol-lowering drugs (CLDs) and ICD-10 Y57.9! collective, respectively. Participants who answered the questionnaire were matched 1:1 with the final EMPAR study data set including TK routine care data and pharmacogenetic data [2]. Thereby, 307 (2.8%) participants without match were excluded. Another 1269 (12%) TK insurees who did not provide fully answered questionnaire scales were also not assessed in this evaluation. The percentage of dropouts in terms of initial male participants (11.2%) was lower than the dropout in female participation (14.7%). Furthermore, 2.4% of the initial population were dropouts without any information on gender due to early exclusion or terminated participation (S1 Table). The total number of the EMPAR questionnaire participants evaluated in terms of their attitude on the basis of the scales of interest was 9323 (88%) (Fig 1). All the following evaluations are performed on the basis of the data sets of this final EMPAR questionnaire population.

With a percentage difference of 72.7%, male participants dominated the survey with respect to women, with 6357 (68.2%) against 2966 (31.8%) (Table 1).

Evaluated participants were 66.2 years old (SD = 11.9) on average with a median age of 68 years. Old patients (> 60 years), middle-aged (36–60 years) and young adults (<36years) represented 71.8% (n = 6694), 25.8% (n = 2402) and 2.4% (n = 227) of the assessed population, respectively (Table 2). Mean age of male and female participants fell within the same age cohort (old patients >60 years), with a mean age of 67.2 (SD = 10.7) years and 64 (SD = 13.7) for male and female participants, respectively. Male participants were on average older than female participants in all three age groups (Table 2).

## Scale score analysis

Internal consistency of the BMQ scales and PSM scale of the EMPAR questionnaire showed values ranging from 0.5–0.8, which falls within a poor to acceptable range. Whereas, the

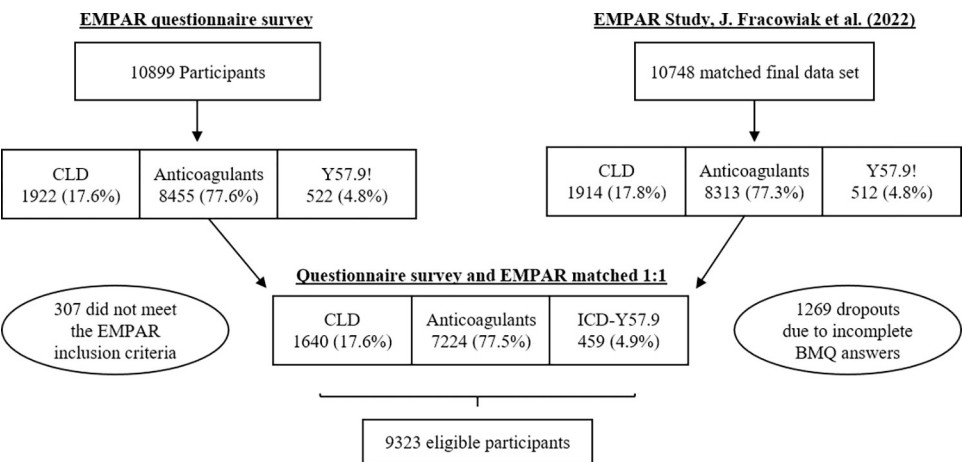

**Fig 1. Enrolment of questionnaire participants: A 1:1 match disqualified patients who did not meet up with the EMPAR enrolment criteria as well as patients who did not submit fully answered scale items.** Anticoagulants: anticoagulant/antiplatelet collective; CLD: cholesterol-lowering drugs; ICD: International Classification of Diseases and Health Related Disorders.

internal consistency of the Natural Remedy scale was unacceptable, with Cronbach's Alpha of 0.2 (Table 3). Results of the Natural Remedy scale were therefore not interpreted and should be treated with caution.

A general overview of the whole questionnaire population showed that the patients obtained moderate scores on the Overuse (mean = 12.1), Harm (mean = 9.2), and PSM (mean = 9.7) scales and a higher score on the Benefit scale (mean = 15.5) with regard to the maximum score of 20 (Table 3). The Gene Testing score in the population was high (mean = 4.1) with regard to the maximum score of 5.

Thus, the questionnaire population showed strongest views in the beneficial aspect of medicines and were moderately concerned with the harm caused by medicines and their overuse, furthermore they also had a moderate perception of their sensitivity towards medicines (Fig 2).

Correlation analysis showed a weak positive correlation between the scales Harm, Overuse and PSM and a weak negative correlation between each of these scales with the Benefit scale with a significance of at least <0.001 (S2 Table).

**Table 1. Age distribution of male vs female participants within each cohort.** The sum total of cohorts shows that the number of male participants was higher than the number of female participants.

| Collective | Gender | Sum | Mean (years) | SD (years) | Median (years) |
|---|---|---|---|---|---|
| **Anticoagulant/antiplatelet drugs** | **Male** | 5193 | 67.9 | 10.6 | 69 |
| | **Female** | 2031 | 65.6 | 13.3 | 68 |
| **CLD** | **Male** | 989 | 64.1 | 10.7 | 64 |
| | **Female** | 651 | 63.6 | 9.9 | 64 |
| **ICD-10 Y57.9!** | **Male** | 175 | 66.1 | 13.4 | 68 |
| | **Female** | 284 | 53.5 | 19.1 | 56 |
| **TOTAL** | **Male** | 6357 | 67.2 | 10.7 | 68 |
| | **Female** | 2966 | 64.0 | 13.7 | 66 |
| | | 9323 | 66.2 | 11.9 | 68 |

CLD: Cholesterol-lowering drugs; SD: standard deviation

**Table 2. Gender and age distribution of evaluated EMPAR participants.**

| Age Range | Sex | N | Mean | SD |
|---|---|---|---|---|
| <36 years (young adults) | Male | 66 | 29.7 | 4.4 |
| | Female | 161 | 27.7 | 4.7 |
| 36 years—60 years | Male | 1583 | 54.0 | 5.0 |
| | Female | 819 | 52.3 | 6.0 |
| >60 years | Male | 4708 | 72.2 | 6.6 |
| | Female | 1986 | 71.8 | 6.6 |
| **Total** | | | | |
| <36 years young adults | | 227 | 28.3 | 4.7 |
| 36 years—60 years | | 2402 | 53.4 | 5.4 |
| >60 years | | 6694 | 72.1 | 6.6 |
| BMQ EMPAR Population | | 9323 | 66.2 | 11.9 |

Descriptive analysis of the BMQ-G12 scales and PSM scales in the different collectives showed that the Cronbach's Alpha values for Harm, Overuse and PSM are relatively consistent across collectives, indicating a sufficient internal consistency (Table 4).

The Benefit scale has a slightly lower Cronbach's Alpha value (poor) but is still applicable. The Natural Remedy scale showed lowest Cronbach's Alpha values, suggesting unacceptable internal reliability, particularly in the anticoagulant/antiplatelet and ICD-10 Y57.9! collectives. Results presented with regard to this scale therefore should be handled with caution.

In the comparison of the scales Overuse, Benefit, PSM, and Natural Remedy in the anticoagulant/antiplatelet vs. the ICD-10 Y57.9! collective, the p-values indicate statistically significant differences between these two groups. In CLD vs ICD-10 Y57.9! for the Overuse, Benefit and PSM scales, the p-values also indicate statistically significant differences (p = <0.001, p = 0.013 and p = <0.001 respectively) (S3 Table). However, there are no statistically significant differences in perceptions in terms of these constructs between the anticoagulant/antiplatelet and CLD collectives. The Harm scale does not exhibit statistically significant differences between any of the compared collectives.

In the scale evaluations, the gender aspect revealed that women scored slightly higher than men in Overuse, Harm and PSM with significant difference in all of these scores. Meanwhile men scored significantly higher in the Benefit scale. (Table 5). However, there is no statistically significant difference in perceptions of Gene Testing between male and female participants.

An overall analysis of the scale scores with a focus on age showed that patients who fell within the range of ≥18 and <36 years (young adults) have a statistically significant stronger belief that doctors overuse prescriptions of drugs in comparison to patients of the groups ≥36 years to ≥60 (middle-aged) years (average score of 12.6 vs.12.1 (p = 0.012)) and patients in the >60years group (old) (average score of 12.6 vs.12.1 (p = 0.011)), respectively (Fig 3).

**Table 3. Means, SDs and Cronbach's alphas of each multi-item scale in the EMPAR questionnaire population (9323 participants).**

| Scales | Mean | SD | Cronbach's Alpha |
|---|---|---|---|
| Harm | 9.2 | 2.5 | 0.63 |
| Overuse | 12.1 | 2.5 | 0.64 |
| Benefit | 15.5 | 2 | 0.5 |
| PSM | 9.7 | 3.1 | 0.8 |
| Natural Remedy | 9.2 | 2.1 | 0.2 |

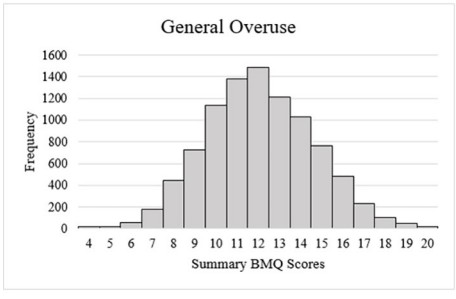
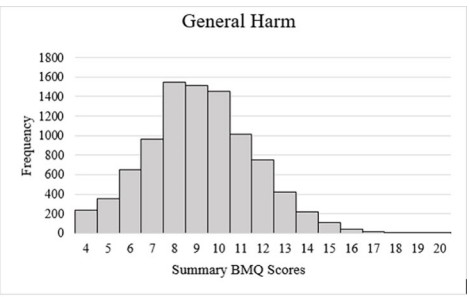
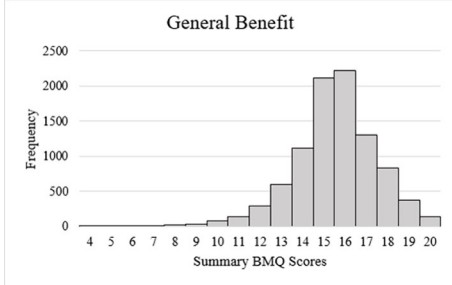
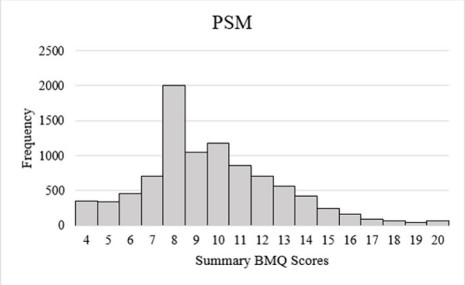

**Fig 2. Distribution of average scores for each subscale of the Beliefs about Medicine Questionnaire-General 12 (Overuse, harm and benefit) and Perceived Sensitivity to Medicines Scale.**

Furthermore, a significance in the difference of PSM scores was detected. Thereby, old patients viewed themselves more often sensitive to the medications they take than young or middle-aged patients (p = 0.01 and p<0.001), respectively (Table 6 and S4 Table). Elderly patients also significantly perceived drugs as more harmful when compared to middle aged patients (p = <0.001). Young adults and middle-aged participants were more open to the idea of getting their genes tested when compared to plus 60-year-old participants (p = 0.03 and <0.001 respectively).

The scale scores of each age group compared across the study collectives, interestingly showed that the young participants of the anticoagulant collective scored higher on the Overuse scale (mean (SD) = 12.7 (2.8)) than middle-aged (mean (SD) = 12.0 (2.6); p = 0.01) and old patients (mean (SD) = 12.0 (2.5); p = 0.002) respectively as shown in Fig 4 and S5 Table.

ICD-10 Y57.9! plus-60-year-olds had a significantly higher Benefit score (mean (SD) = 15.9 (2.0)) than the same age group in the anticoagulant/antiplatelet (mean (SD) = 15.5 (2.0); p = 0.01) and CLD collective (mean (SD) = 15.5 (2.0); p = 0.006) respectively (S5 Table). They furthermore significantly perceived themselves as more sensitive to drugs (PSM mean (SD) = 10.4 (3.1)) compared to the same age group of the anticoagulant/antiplatelet collective (PSM

**Table 4. Means, SDs and Cronbach's alphas of each scale in the EMPAR questionnaire collectives.**

| Collectives | BMQ-G12 | | | | | | | | | | | | | | | | |
| | | Harm | | | Overuse | | | Benefit | | | PSM | | | Natural Remedy | | |
| | Sum | Mean | SD | Cronbach's Alpha | Mean | SD | Cronbach's Alpha | Mean | SD | Cronbach's Alpha | Mean | SD | Cronbach's Alpha | Mean | SD | Cronbach's Alpha |
|---|---|---|---|---|---|---|---|---|---|---|---|---|---|---|---|---|
| anticoagulant/antiplatelet | 7224 | 9.2 | 2.4 | 0.63 | 12.0 | 2.5 | 0.64 | 15.5 | 2 | 0.53 | 9.7 | 3 | 0.8 | 9.2 | 2 | 0.3 |
| CLDs | 1640 | 9.2 | 2.5 | 0.62 | 12.1 | 2.5 | 0.6 | 15.5 | 2 | 0.52 | 9.7 | 3.2 | 0.82 | 9.3 | 2 | 0.14 |
| ICD-10 Y57.9! | 459 | 9.4 | 2.5 | 0.61 | 12.6 | 2.5 | 0.63 | 15.8 | 1.8 | 0.5 | 10.3 | 3.3 | 0.81 | 9.4 | 2.2 | 0.05 |

**Table 5. Differences in perceptions between male and female participants in terms of the evaluated scale scores (2-sided t-test).**

| | | Overuse | Harm | Benefit | PSM | Natural Remedy | Gene Testing |
|---|---|---|---|---|---|---|---|
| | N | Mean (SD) | Mean (SD) | Mean (SD) | Mean (SD) | Mean (SD) | Mean (SD) |
| **Male** | 6357 | 12 (2.5) | 9.2 (2.4) | 15.6 (2)* | 9.3 (2.8) | 9.1 (2) | 4.1 (0.9) |
| **Female** | 2966 | 12.2 (2.6)*** | 9.3 (2.5)* | 15.5 (2) | 10.6 (3.5)*** | 9.3 (2.2)*** | 4.1 (1) |
| **p-value** | | <0.001 | 0.04 | 0.043 | <0.001 | <0.001 | 0.98 |

*** = p < 0.001

** = p < 0.01

* = p < 0.05

mean (SD) = 9.8 (3.0); p = 0.04) and CLD (PSM mean (SD) = 9.8 (3.1), p = 0.03) and had a significantly higher Overuse score (mean (SD) = 12.7 (2.3) compared to the same age groups in the other two collectives (anticoagulated/antiplatelet: mean (SD) = 12.0 (2.5); p <0.001 and CLD: mean (SD) = 12.2 (2.5); p-value = 0.002).

Middle-aged ICD-10 Y57.9! patients significantly considered themselves as more sensitive (PSM mean (SD) = 10.7(3.5)) to drugs compared to middle-aged patients in the anticoagulant/antiplatelet group (mean (SD) = 9.3 (3.1); p<0.001) as well as in the CLD group (mean (SD) = 9.5 (3.2); p<0.001). There was no difference between the collectives anticoagulant/antiplatelet and cholesterol lowering drugs.

In all age groups across all collectives, participants leaned more towards carrying out gene testing with all groups having a similar Gene Testing score (Fig 4).

In a comparison of the attitudes of male and female participants of different age groups, only the young male participants (highest age/gender group score) showed a significantly higher Benefit score than female participants which was comparable to the gender evaluation within the entire population, where male participants scored higher than female participants. However, in the higher age groups no significant difference was observed between male and female participants (Table 7).

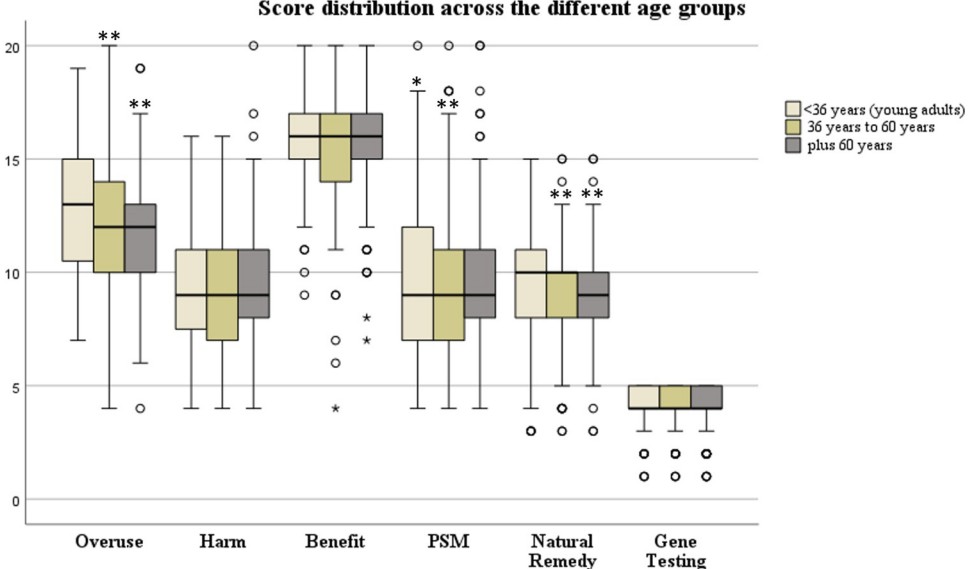

**Fig 3. Distribution of BMQ scores, PSM, Natural Remedy and Gene Testing scores across the different age groups in the EMPAR population.**

**Table 6. Analysis of the scale scores between the different age groups of the evaluated EMPAR population.** (ANOVA and Tukey's HSD test).

|  | Overuse Mean (SD) | Harm Mean (SD) | Benefit Mean (SD) | PSM Mean (SD) | Natural Remedy Mean (SD) | Gene testing Mean (SD) |
|---|---|---|---|---|---|---|
| <36 years | 12.6 (2.9) | 9.3 (2.6) | 15.7 (2.0) | 9.2 (3.4)** | 9.6 (2.4) | 4.2 (0.8) |
| 36–60 years | 12.1 (2.6)* | 9 (2.5)* | 15.5 (2.1) | 9.4 (3.2)*** | 9.2 (2.1)** | 4.1 (0.9) |
| >60years | 12.1 (2.5)* | 9.3 (2.4) | 15.5 (2.0) | 9.8 (3.0) | 9.2 (2.1)** | 4.1 (1)*/*** |

Overuse, PSM and Natural Remedies scores of female participants were significantly higher in the middle and old age group than of male participants. However, the highest mean Overuse score was achieved by young female participants and the highest PSM score was detected for female participants of the old age group.

## Genotype and phenotype evaluations in terms of patient perception

All EMPAR participants gave their written consent to being genotyped for specific pharmacogenes, as described by Fracowiak et al. (2022), consequently most participants answered the question about genetic testing with high agreement (mean score (SD) = 4.1 (0.89)). A general analysis showed that in the EMPAR questionnaire population young adults scored slightly more positive in terms of the Genetic Testing scale compared to middle-aged and old participants (Table 7). In the anticoagulant/antiplatelet collective, participants' perception to drug therapy was analyzed with regard to the proportion of actionable variants listed and evaluated previously [2]. These variants were relevant for the response to therapy with anticoagulant/antiplatelet drugs of study interest. Patients with 0, 1, 2, 3, 4 and ≥5 of the investigated actionable variants showed no significant difference in the evaluated scale scores. A tendency for a higher Harm, Overuse, Natural Remedy and PSM score was identified for patients with ≥5 actionable variants, however, here sample size was low with 18 participants affected (Fig 5).

A general analysis of scale scores of questionnaire participants stratified by the different phenotypes of the pharmacogenes *CYP2C19* and *CYP2C9*, did not result in significant differences for any of the pharmacogenes of interest. An analysis of anticoagulant/antiplatelet collective patients with clopidogrel or clopidogrel + acetylsalicylic acid (ASA) prescriptions, revealed that patients of the clopidogrel prescription subgroup scored a higher PSM (mean (SD) = 9.9 (3.2); p = 0.003) vs those with no clopidogrel prescriptions in the evaluated period. Furthermore, patients with clopidogrel + ASA prescriptions had significantly higher Harm scale sores (mean (SD) = 9.3 (2.5), p = 0.001) than patients who did not receive these prescriptions in the evaluated period (Table 8).

In previous pharmacoepidemiological analyses on the basis of conditional logistic regression models, it was identified that CYP2C19 ultra rapid metabolism in clopidogrel treatment was associated with increased hemorrhagic events in the EMPAR population [28]. Such events may have influenced the patients' attitude towards medicines. We therefore performed further scale score evaluations in the subgroups of EMPAR questionnaire participants who were prescribed clopidogrel or clopidogrel with ASA to evaluate the attitude towards medicines in conjunction with phenotypes of the relevant clopidogrel metabolizing CYP450 enzymes in these groups. Thereby, the focus was on extreme phenotypes (ultra-rapid metabolizer (UM) or poor metabolizer- (PM)) against normal metabolizers (NM) in CYP2C19 and additionally CYP2C9 (NM vs PM) which are involved in clopidogrel metabolism (Table 9).

However, no significant difference in the mean scale scores of the analyzed medication subgroups was registered between CYP2C19 UM and NM or between CYP2C19 and CYP2C9 extreme and normal phenotypes in general.

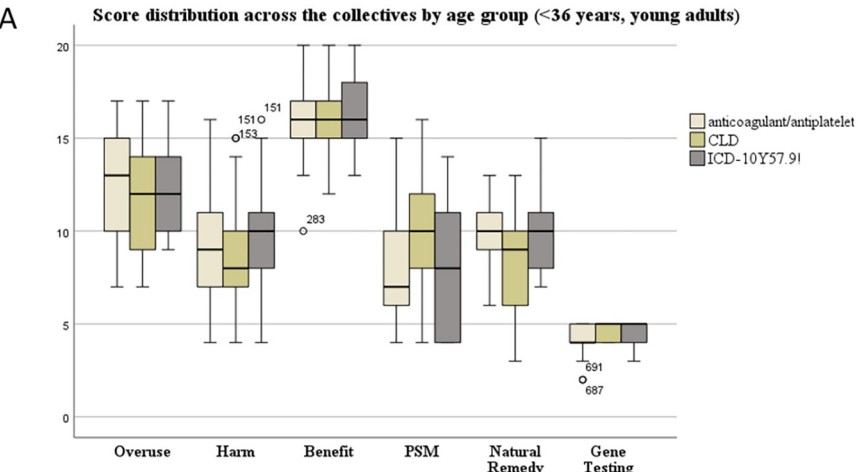

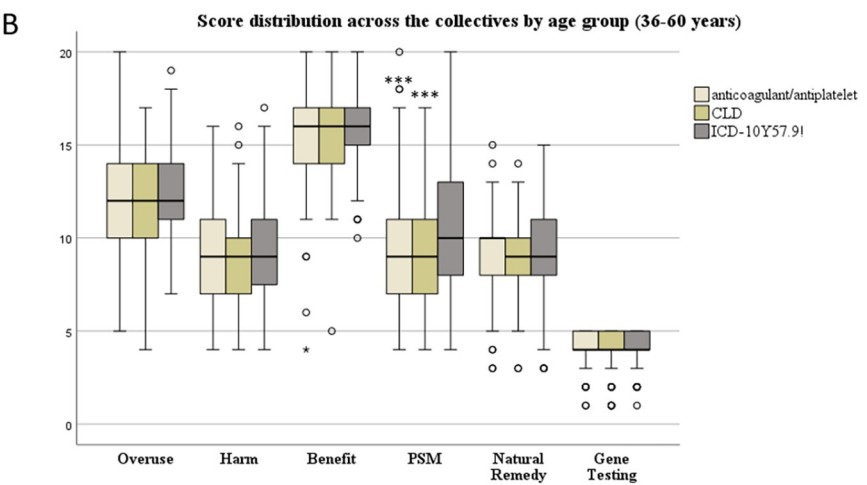

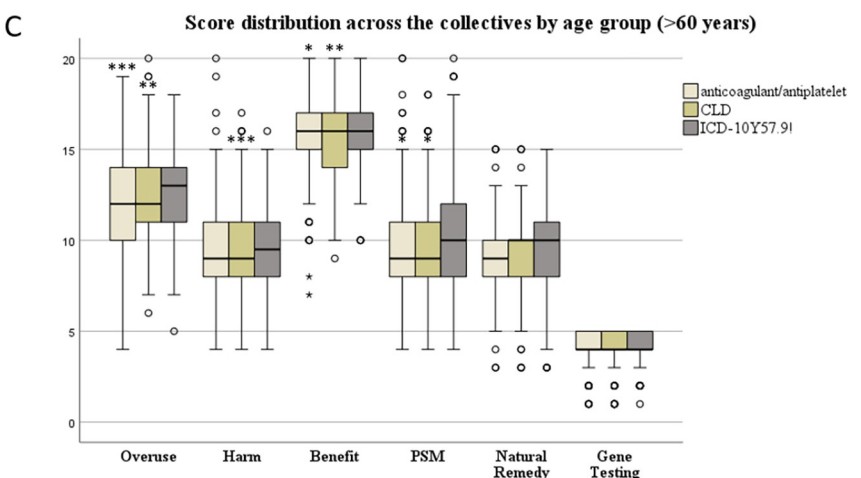

**Fig 4. Distribution of BMQ-General 12, PSM, Natural Remedy and Gene Testing scale scores showing the outcomes associated with age across the collectives.**

## Discussion

The present study aimed to investigate patients' beliefs about medicines in the context of the German EMPAR study which also evaluated the potential of pharmacogenetic testing of metabolic profiles in routine clinical care. The EMPAR questionnaire was utilized to assess patients' views on medicines, natural remedies, pharmacogenetic testing and patient's sensitivity to medicines. The study population included patients with initial prescriptions of anticoagulants or antiplatelets, cholesterol-lowering drugs, and those with complications due to medicines or drugs in 2014–2017 [1, 2].

The results revealed that patients generally held moderate views on the overuse and harm of medicines, with stronger beliefs in the benefits of pharmaceuticals and lesser concern regarding their sensitivity to medicines. This finding aligns with previous literature which shows that more substantial views on benefit will bring about weaker perceptions of the overuse and harmfulness of medicines and vice versa [7, 29].

Patients of the ICD-10 Y57.9! collective, expressed more negative views on the overuse of medications and perceived themselves as more sensitive to drugs than the other two collectives. This observation was not unexpected as the patients in the ICD-10 Y57.9! collective experienced some form of complication due to one or more prescribed or taken medicines or drugs. Similar findings were also observed by De Smedt et al. (2011) who revealed that negative medication beliefs are associated with experienced adverse drug events in patients with heart failure [30]. Furthermore, consistent to previous observations, also in the EMPAR population Perceived Sensitivity to Medicines score levels correlated with negative beliefs about drugs (Overuse and Harm scores). However, in the ICD-10 Y57.9! collective the proportion of female participants was higher than that of male participants in contrast to the other two EMPAR questionnaire collectives. The observed association thus may be influenced by gender.

Gender also appeared to play a role in patients' perceptions, as women expressed more concern about overuse, harm, and their sensitivity to medicines compared to men. On the other hand, men perceive medicines as more beneficial. This gender difference in beliefs about medicines is consistent with previous research and may be attributed to various social and cultural factors [31]. A few studies examining the impact of gender and age on drug prescription in primary care have demonstrated that these factors also influence prescription patterns [32–35].

**Table 7. Comparison of scale scores between male and female participants within the evaluated age groups (two tailed t-test).**

| Scale | | Age group comparison of male and female participants | | | | | |
| --- | --- | --- | --- | --- | --- | --- | --- |
| | | <36 years | | 36years- 60years | | >60years | |
| | | Male | Female | Male | Female | Male | Female |
| | N | 66 | 161 | 1583 | 819 | 4708 | 1986 |
| Overuse | Mean (SD) | 12 (2.9) | 12.8 (2.9) | 12 (2.6) | 12.2 (2.6)* | 12 (2.5) | 12.2 (2.5)** |
| | pValue | | | | 0.012 | | 0.007 |
| Harm | Mean (SD) | 8.8 (2.8) | 9.5 (2.6) | 8.9 (2.5) | 9.1 (2.6) | 9.3 (2.4) | 9.3 (2.4) |
| Benefit | Mean (SD) | 16.2* (2.1) | 15.5 (1.9) | 15.5 (2.1) | 15.4 (2.1) | 15.5 (2.0) | 15.5 (1.9) |
| PSM | mean (SD) | 8.6 (3.4) | 9.4 (3.5) | 9 (2.8) | 10.3 (3.6)*** | 9.4 (2.7) | 10.8*** (3.5) |
| Natural Remedy | Mean (SD) | 9.3 (2.2) | 9.8 (2.6) | 9.1 (2.1) | 9.3* (2.2) | 9.2 (2) | 9.3* (2.1) |
| Gene Test | Mean (SD) | 4.3 (0.7) | 4.2 (0.9) | 4.2 (0.9) | 4.1 (0.9) | 4.1 (0.9) | 4.1 (0.9) |

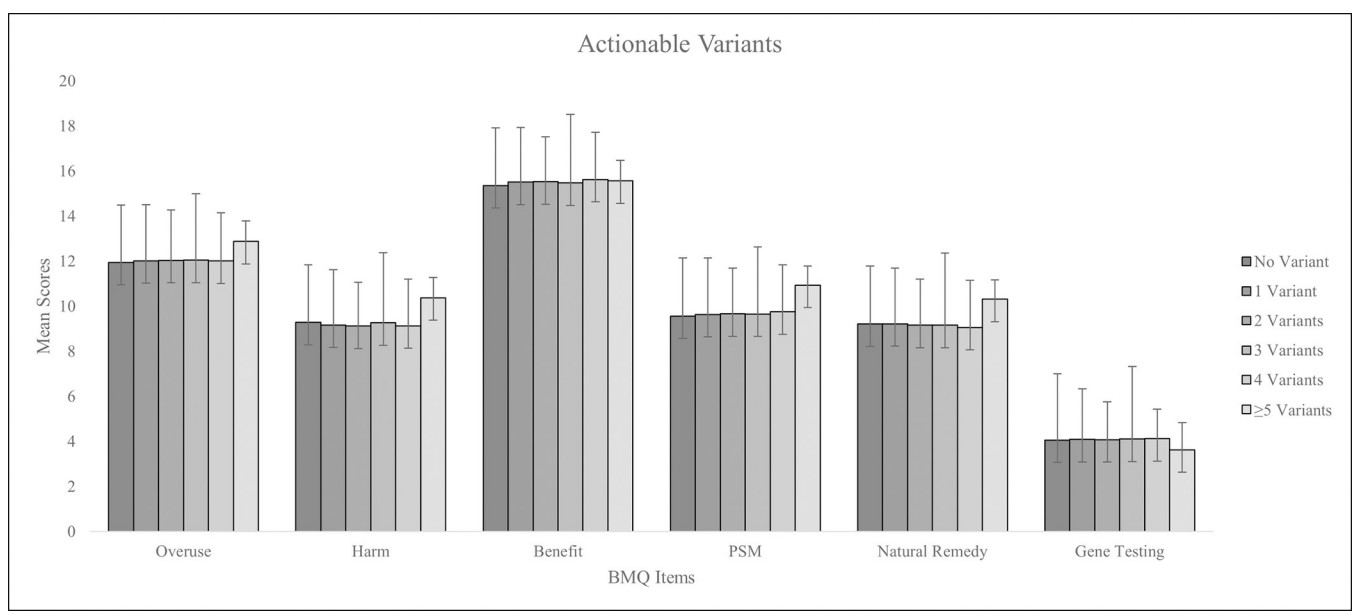

**Fig 5. Assessment of BMQ, PSM, Natural Remedy and Gene Testing scale scores with respect to the number of actionable variants per patient in the EMPAR questionnaire population.**

Furthermore, according to several studies, women use medications on average more frequently than men [31, 35, 36]. Unfortunately, in clinical trials for many drugs, enrolment of women has not occurred sufficiently which often leads to the same dose prescriptions for male and female patients in clinical practice [37, 38]. It is therefore possible that female patients are routinely overmedicated [37]. Furthermore, for several drug groups, including different cardiovascular drug groups, ADRs were reported to be more frequent in women than men [39–41]. Indeed, in the pharmacokinetics of a variety of drugs, sex differences were reported. Women were reported to be exposed to a higher drug concentration in the blood upon

**Table 8. Assessment of BMQ, PSM, Natural Remedy and Gene Testing in EMPAR questionnaire patients of the anticoagulant/antiplatelet collective undergoing treatment with clopidogrel (two-sided t-test).**

|  |  | No Clopidogrel | Clopidogrel | No Clopidogrel+ASA | Clopidogrel+ASA |
|---|---|---|---|---|---|
| Overuse | N | 5987 | 515 | 5044 | 1458 |
|  | Mean | 11.9 | 11.9 | 11.9 | 11.9 |
|  | SD | 2.7 | 2.5 | 2.7 | 2.6 |
| Harm | Mean | 9.1 | 8.9 | 9.0 | 9.3*** |
|  | SD | 2.5 | 2.5 | 2.5 | 2.5 |
| Benefit | Mean | 15.3 | 15.3 | 15.3 | 15.3 |
|  | SD | 2.3 | 2.1 | 2.2 | 2.3 |
| PSM | Mean | 9.6 | 9.9*** | 9.6 | 9.6 |
|  | SD | 3.1 | 3.2 | 3.1 | 3.0 |
| Natural Remedy | Mean | 9.2 | 9.0 | 9.1 | 9.3*** |
|  | SD | 2.1 | 2.2 | 2.2 | 2.1 |
| Gene Testing | Mean | 4.1 | 4.1 | 4.1 | 4.1 |
|  | SD | 0.9 | 0.9 | 0.9 | 0.9 |

*** $p < 0.001$ (2-sided t-test)

**Table 9. Beliefs about Medicine Questionnaire, PSM, Natural Remedy and Gene Testing scale evaluation in patients of the clopidogrel and clopidogrel with acetyl-salicylic acid prescription subgroups in conjunction with CYP2C9 and CYP2C19 metabolizer phenotypes (two-sided t-test or one-way ANOVA).**

| Clopidogrel | | | | | | | Clopidogrel with ASA | | | | |
|---|---|---|---|---|---|---|---|---|---|---|---|
| | | CYP2C9 | | CYP2C19 | | | CYP2C9 | | CYP2C19 | | |
| | | NM | PM | NM | PM | UM | NM | PM | NM | PM | UM |
| **Overuse** | N | 3223 | 198 | 2061 | 116 | 253 | 1388 | 100 | 856 | 56 | 112 |
| | Mean | 12.03 | 12.01 | 12.04 | 11.68 | 11.88 | 12.00 | 12.48 | 12.12 | 11.64 | 11.94 |
| | SD | 2.6 | 2.6 | 2.6 | 2.7 | 2.8 | 2.5 | 2.7 | 2.5 | 2.1 | 2.6 |
| **Harm** | Mean | 9.17 | 9.22 | 9.18 | 8.80 | 8.92 | 9.22 | 9.62 | 9.22 | 9.07 | 9.08 |
| | SD | 2.4 | 2.3 | 2.5 | 2.4 | 2.4 | 2.4 | 2.8 | 2.5 | 2.4 | 2.4 |
| **Benefit** | Mean | 15.52 | 15.70 | 15.60 | 15.78 | 15.58 | 15.44 | 15.53 | 15.48 | 15.86 | 15.46 |
| | SD | 2.0 | 2 | 2 | 2.2 | 2 | 2.1 | 2.1 | 2.0 | 1.6 | 1 |
| **PSM** | Mean | 9.65 | 9.45 | 9.54 | 9.63 | 9.38 | 9.73 | 9.89 | 9.73 | 9.48 | 9.62 |
| | SD | 3.1 | 2.9 | 3.1 | 3.1 | 3.0 | 3.0 | 3.0 | 3.0 | 3.0 | 2.8 |
| **Natural Remedy** | Mean | 9.17 | 9.11 | 9.14 | 9.05 | 8.83 | 9.34 | 9.41 | 9.21 | 9.20 | 9.17 |
| | SD | 2.1 | 2.1 | 2.1 | 2.0 | 2.3 | 2.0 | 2.1 | 2.6 | 1.9 | 1.9 |
| **Gene Testing** | Mean | 4.08 | 4.10 | 4.08 | 4.13 | 4.06 | 4.11 | 3.98 | 4.10 | 4.16 | 4.09 |
| | SD | 0.9 | 0.8 | 0.9 | 0.9 | 0.9 | 0.9 | 1.0 | 0.9 | 0.9 | 1.0 |

NM: normal metabolizer; PM: poor metabolizer; UM: ultra-rapid metabolizer. No significant differences were observed.

standard dose and longer elimination times which could not be explained by weight differences only [38]. Therefore, it can be assumed that women may be more susceptible to the harmful effects and adverse reactions caused by medicines leading their perception to be more sensitive towards drugs. However, a stratified analysis by age showed that, significant differences between male and female participants in terms of the Overuse and PSM score were identified only in the middle-aged and elderly although young women had the highest Overuse scores. Benefit scores were significantly higher only in the young male group (highest score) compared to female participants of the same age group.

Thus, age was another factor influencing patients' beliefs. Older patients (>60 years) considered themselves more sensitive to medicines which might be related to the higher likelihood of experiencing adverse drug reactions in older adults due to factors like frailty and multi-morbidity [42, 43]. ADRs and multimorbidity in older persons were shown to be related. Other studies have demonstrated that the probability of adverse drug reactions rises with the number of chronic conditions which can occur as a result of different interactions e.g., when a medication intended to treat one ailment worsens the symptoms or signs of a different underlying disorder [42–44]. Other studies also show that progressive age-related decline in physiological systems also known as "frailty" increases the risks of ADRs [45]. As noted by Muhlack D.C et al. (2018), older persons with frailty, co-morbidity, and functional impairment had a higher risk of receiving a potentially inappropriate medication prescription. The longer treatment history thus may be more often affected by negative treatment experiences that can induce negative carry-over effects and may impact also future treatment response [5]. Therefore, extra caution is necessary in prescribing medications to these patients as they are more likely to experience adverse reactions which may impact their attitude towards therapy and also future compliance to therapy [46]. The above-mentioned reasons may also explain why old patients (>60 years), especially from the ICD-10 Y57.9! and from the anticoagulant/antiplatelet collective, which reflect more serious health issues related to ADRs or frailty respectively, had higher PSM scores than patients of the cholesterol-lowering drugs collective.

The patients' perspective on medicines on the basis of education levels was assessed. However, as mainly no information on the education level was available and predominantly high education levels were reported in the study population (S6 Table), the results were inconclusive and therefore not included in the manuscript. The study also explored patients' willingness to participate in genetic testing, with most participants, especially young participants, showing a positive attitude towards the procedure. However, taking into account that the study's inclusion criteria required the provision of DNA samples [1, 2], potential bias in the responses must be considered. Furthermore, score measurements on the basis of only one question tend to be less valid and accurate since individual items usually have random measurement errors which can be averaged out if individual item scores are summed to obtain a total score [47]. In addition, individual scores cannot discriminate among fine degrees and they very unlikely represent a fully theoretical concept [48]. The evaluation of the Gene Testing item score was still considered to receive an impression of the preference of study participants with regard to pharmacogenetic guidance of their therapy dependent on age, gender and pharmacogenetic profile.

Regarding genetic factors, the presence of extreme phenotypes or actionable variants did not result in significant differences in patients' attitudes compared to patients without extreme phenotypes or actionable variants. However, a tendency of a higher Harm, Overuse and PSM scores was observed for a small participant group with ≥5 actionable variants. Probably, due to a subjective experience in terms of medication use [49], carriers of lower numbers of actionable variants are not able to perceive such pharmacogenetic associations, unless they are strong or may be linked more often with problematic effects after medication intake. The phenotype of single pharmacogenes such as *CYP2C9* and *CYP2C19* showed no significant impact on patient's scale scores.

In a more detailed analysis with regard to specific medicines, patients under clopidogrel treatment showed significantly higher PSM scores and patients under clopidogrel + ASA treatment showed a significantly higher Harm scale score compared to patients without these prescriptions in the evaluated period (Table 8). However, in these medication subgroups score differences were not observed when patients were analysed further, stratified in terms of relevant phenotypes such as of the clopidogrel metabolizing CYP2C9 and CYP2C19 (Table 9).

Overall, the study provides valuable insights into patients' beliefs about medicines and their perceptions of drug therapy in the context of a pharmacogenetic background. Furthermore, it sheds light on the importance of age, gender, medication and previous experiences of complications due to medicines or drugs in shaping patients' beliefs and perceptions of drug therapy. However, differences in perceptions across EMPAR collectives, gender, age, and medication subgroups were low. Significant associations still could be observed due to large sample sizes.

Understanding patients' beliefs about medicines and their attitude towards drug therapy can aid healthcare professionals in tailoring treatments to individual needs and preferences. Likewise, addressing patients' concerns about treatment and providing appropriate education can enhance medication adherence and overall treatment success. The influence of genetic factors on patients' perceptions of drug therapy is an essential aspect to consider as individual genetic variability can significantly impact drug response and safety [3]. Thus, the integration of pharmacogenomics and personalized medicine, taking into account also patient's concerns about the treatment, may have potential to optimize drug therapy, minimize adverse drug reactions, and improve patient outcomes and adherence.

## Conclusion

In the EMPAR study, age was associated with a higher impact on how vulnerable participants perceived themselves to medicines compared to gender and pharmacogenetic aspects,

especially if they have already experienced at least one complication due to medicines or drugs. However, age is the only variable that changed over time for the participants in this evaluation and may be stronger accompanied with a changing body experience related to the daily use of medications. Gender showed an impact while pharmacogenetic composition, which does not change in the course of a life time, had almost no perceived impact measured by BMQ-12 and PSM. However, participant's subjective attitude towards medicines was influenced by all of these evaluated variables to some extent. The present BMQ and the additionally applied question blocks in the EMPAR questionnaire may not be a method sensitive enough to distinguish whether e.g. the pharmacogenomic background is associated with patient's perception on vulnerability and attitude towards medicines. Devising further methods through which a patient's perception of medicinal therapy can be assessed in the context of the pharmacogenetic composition would be of additional value as this may better resolve and significantly strengthen and validate the present BMQ and PSM score results.

## Supporting information

**S1 Table. The distribution of male and female participation and dropouts.** Patients who were excluded from the BMQ evaluation either did not fulfil the necessary inclusion criteria, had incomplete BMQ answers which could not be evaluated or IDs could not be identified within the EMPAR population due to terminated participations.
(PDF)

**S2 Table. Correlation analysis in the BMQ-General 12 and PSM.** Correlation coefficients are shown for each analysis. * The correlation is significant at the 0.01 level (2-sided t-test).
(PDF)

**S3 Table. Multiple comparisons of the collectives with Tukey's Honest Significant Difference analysis (HSD).**
(PDF)

**S4 Table. Tukey's HSD showing multiple comparisons across the age ranges and the BMQ, PSM, Natural Remedy and Gene Testing scales.**
(PDF)

**S5 Table. Association of age with patients' perception of medicine within the collectives (2-sided t-test).** P-values were calculated within each age group with respect to the collectives. *** = p < 0.001; ** = p < 0.01; * = p < 0.5.
(PDF)

**S6 Table.**
(XLSX)

## Acknowledgments

The authors thank Daria Langner, Marco Garling and Felix Falkenberg for the coordination of the recruitment and selection of TK insurees according to study criteria. Furthermore, the authors thank Steffen Heß for quality control of participants' routine care data and Jochen Fracowiak and Kathrin Worm for questionnaire data management.

## Author Contributions

**Conceptualization:** Tatjana Huebner, Michael Steffens, Britta Haenisch, Julia Stingl, Roland Linder, Catharina Scholl.

**Formal analysis:** Veronica Atemnkeng Ntam.

**Funding acquisition:** Michael Steffens, Britta Haenisch, Julia Stingl, Roland Linder.

**Methodology:** Michael Steffens, Britta Haenisch, Julia Stingl, Roland Linder.

**Project administration:** Tatjana Huebner, Michael Steffens.

**Resources:** Michael Steffens, Christoph Roethlein.

**Supervision:** Tatjana Huebner, Michael Steffens, Catharina Scholl.

**Visualization:** Veronica Atemnkeng Ntam.

**Writing – original draft:** Veronica Atemnkeng Ntam, Tatjana Huebner, Catharina Scholl.

**Writing – review & editing:** Veronica Atemnkeng Ntam, Tatjana Huebner, Michael Steffens, Christoph Roethlein, Britta Haenisch, Julia Stingl, Roland Linder, Catharina Scholl.

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
