## [Decision Letter · Decision Letter 0]

23 Jun 2024

PONE-D-24-09164The influence of age, gender and pharmacogenetic profiles on the beliefs about medicines in the German EMPAR studyPLOS ONE

Dear Dr. Huebner,

Thank you for submitting your manuscript to PLOS ONE. After careful consideration, we feel that it has merit but does not fully meet PLOS ONE’s publication criteria as it currently stands. Therefore, we invite you to submit a revised version of the manuscript that addresses the points raised during the review process.

**ACADEMIC EDITOR: ** Overall, the reviewers are pleased with the work submitted. A slight concern pertaining the research question, being the link of Pgx markers and belives towards medicine is vague and should be clarified further, focusing more on the impact of having pgx tests and their outlook on medicine afterwards would be more helpful.

We look forward to receiving your revised manuscript.

Kind regards,

Hoh Boon-Peng, PhD

Academic Editor

PLOS ONE

“This EMPAR study was funded by the Innovation Fund of the Federal Joint Committee in Germany (Gemeinsamer Bundesausschuss). Grant number: 01VSF16047.”

Reviewers' comments:

Reviewer's Responses to Questions

**Comments to the Author**

1. Is the manuscript technically sound, and do the data support the conclusions?

Reviewer #1: Yes

Reviewer #2: Yes

2. Has the statistical analysis been performed appropriately and rigorously? 

Reviewer #1: Yes

Reviewer #2: Yes

3. Have the authors made all data underlying the findings in their manuscript fully available?

Reviewer #1: Yes

Reviewer #2: Yes

4. Is the manuscript presented in an intelligible fashion and written in standard English?

Reviewer #1: Yes

Reviewer #2: Yes

5. Review Comments to the Author

Reviewer #1: Dear Authors,

Congratulations on your study. some of my comment are as follows:

1. In the title, the word outlook or perspective would better represent views and beliefs.

2. How far back does the retropective data collected?

3. EMPAR questionnaire: Would be great to have a simple explanation on the questionnaire, such as how many questions for each block. for the gene testing block, is one question enough to understand the participants' prespective?

4. Figure 3: I find the box plot contains less information than the suppl. table 4. I prefer the table to be in the main manuscript.

5. From the ICD-10 Y57.9! database, does any of the subjects benefited from genotyping? what were their score for gene testing compared to the rest of the population?

6. Figure 6 carry very little information on its own. is there correlating data from their ICD code, for example, does the ICD code relate to the variant(s)? does genetic testing solve their ICD code issue?

Reviewer #2: This manuscript provides valuable insights into the factors influencing beliefs about medications. All the analyses were well conducted. Following are some minor comments:

1. There should be an explanation of “BMQ” when it is first mentioned in the main context (line 35). I suggest a thorough proofreading.

2. Did the authors considered genetic variations on other genes besides the 4 genes mentioned in the manuscript?

3. There might be additional interesting findings by analyzing a) the association between education levels and patients' beliefs and b) the interaction between different variables.

6. PLOS authors have the option to publish the peer review history of their article (what does this mean?). If published, this will include your full peer review and any attached files.

Reviewer #1: No

Reviewer #2: No

---

## [Author Response · Author response to Decision Letter 0]

30 Aug 2024

Dear reviewers,

dear editor,

Thank you very much for your comments, suggestions and questions to improve the quality of the manuscript of our research article! Please find our answers and descriptions of implemented changes below.

Thank you very much, we checked the requirements and adjusted the manuscript accordingly!

“This EMPAR study was funded by the Innovation Fund of the Federal Joint Committee in Germany (Gemeinsamer Bundesausschuss). Grant number: 01VSF16047.”

We included the suggested statement "The funders had no role in study design, data collection and analysis, decision to publish, or preparation of the manuscript." according to the instructions. Further amendments were not necessary.

Thank you very much, we prepared the data table S2 Table1 that will be available in the supporting information. We furthermore changed the Email address of the corresponding author, as the old address will soon not be valid.

We performed changes to the manuscript according to the instructions.

We performed changes to the manuscript according to the instructions.

We revised the manuscript according to the instructions.

Reviewers' comments:

Reviewer's Responses to Questions

Reviewer #1: Dear Authors,

Congratulations on your study. some of my comment are as follows:

1. In the title, the word outlook or perspective would better represent views and beliefs.

Thank you very much! We changed the wording accordingly. The new title of the manuscript is “The influence of age, gender and pharmacogenetic profiles on the perspective on medicines in the German EMPAR study”

2. How far back does the retrospective data collected?

Data of a range of 3-4 years was collected for each participant. Data of one year prior to initial prescription of the drugs of interest or the Y57.9! diagnosis, the year of an initial prescription or the Y57.9! diagnosis if it was detected in 2013-2018 and data of one to two subsequent years (for recruited participants in 2018 only data of one subsequent year was available) was provided by the involved health insurance company. Thus, the earliest data was available from 2012 and the latest time at which data was retrieved was 2019. We included the information in the methods section in line 132-135 of the revised version.

3. EMPAR questionnaire: Would be great to have a simple explanation on the questionnaire, such as how many questions for each block. for the gene testing block, is one question enough to understand the participants' prespective?

We adjusted the explanation of the questionnaire in the methods section in line 155-157 of the revised version to improve the information on the number of questions in each block. The gene testing question was not part of a validated questionnaire block such as the Harm, Benefit, Overuse and Perceived Sensitivity to Medicines blocks of the validated BMQ questionnaire. The one question item on gene testing is not sufficient for a validation and thorough assessment of the participants’ perspectives on gene testing. We included it to receive an impression of the participants’ attitude towards the use of preemptive testing in routine care. However, for an appropriate evaluation a different study design and an appropriate, validated question block is necessary as here participants explicitly gave their consent to participate in gene testing. These limitations were addressed in the discussion in line 462-471.

4. Figure 3: I find the box plot contains less information than the suppl. table 4. I prefer the table to be in the main manuscript.

Thank you very much! The table is indeed more informative. We removed the box plot and exchanged it with the table. The manuscript and the supplementary material were adjusted accordingly.

5. From the ICD-10 Y57.9! database, does any of the subjects benefited from genotyping? what were their score for gene testing compared to the rest of the population?

The occurrence and frequency of the ICD-10 Y57.9! diagnosis is currently evaluated in terms of a correlation with specific prescriptions and gene variants or phenotypes. The results will be published elsewhere. The score for gene testing in the ICD-10 Y57.9! collective did not differ significantly compared to the other collectives of the EMPAR population.

6. Figure 6 carry very little information on its own. is there correlating data from their ICD code, for example, does the ICD code relate to the variant(s)? does genetic testing solve their ICD code issue?

Figure 6. only shows how the number or accumulation of actionable variants correlates with the evaluated questionnaire scores. Further evaluations in terms of specific ICD 10 codes, medications and relevant actionable variants or phenotypes are currently in progress and will be published elsewhere. 

Reviewer #2: This manuscript provides valuable insights into the factors influencing beliefs about medications. All the analyses were well conducted. Following are some minor comments:

1. There should be an explanation of “BMQ” when it is first mentioned in the main context (line 35). I suggest a thorough proofreading. an explanation of “BMQ” when it is first mentioned in the main context (line 35)

We performed a thorough proofreading and checked all abbreviations as suggested. An explanation of “BMQ” was added in line 34 in the revised version when it is first mentioned in the main context. An explanation of “EMPAR” was added in line 27-29, of “ICD” in line 41, of “CYP” in line 193, of “SLCO1B1”in line 195, of “VKORC1” in line 196 of the revised version when it is first mentioned in the main context. No further unexplained abbreviations at first mentioning were detected.

2. Did the authors considered genetic variations on other genes besides the 4 genes mentioned in the manuscript?

In this manuscript we only focused on the genes and gene variants with the highest evidence (PharmGKB Clinical Annotation Levels of Evidence 1A or 1B) to be actionable with regard to the medication of main study interest (cholesterol lowering drugs and anticoagulants/ antiplatelet agents). Further evaluations including comedications and other relevant genes/gene variants covered by the iPLEX® PGx 74 and the VeriDose® CYP2D6 CNV panel are in progress and will be published elsewhere.

3. There might be additional interesting findings by analyzing a) the association between education levels and patients' beliefs and b) the interaction between different variables.

Thank you very much! The evaluation of the suggested issues is indeed interesting. The patients' beliefs with regard to education levels were assessed; however, as mainly no information on the education level was available and predominantly high education levels were reported in the study population, the results were inconclusive and therefore not included in the manuscript. We included the information on the education level in the discussion section in line 457-460 of the revised version. 

The interaction between different variables is currently assessed in terms of specific prescriptions, pharmacogenes and ICD-10 codes and will be published elsewhere. 

Thank you for your consideration! 

With best regards,

Dr. Tatjana Hübner

---

## [Editor Report · Decision Letter 1]

17 Sep 2024

The influence of age, gender and pharmacogenetic profiles on the perspective on medicines in the German EMPAR study

PONE-D-24-09164R1

Dear Dr. Huebner,

We’re pleased to inform you that your manuscript has been judged scientifically suitable for publication and will be formally accepted for publication once it meets all outstanding technical requirements.

Kind regards,

Hoh Boon-Peng, PhD

Academic Editor

PLOS ONE
---

## [Editor Report · Acceptance letter]

30 Sep 2024

PONE-D-24-09164R1 

PLOS ONE

Dear Dr. Huebner, 

I'm pleased to inform you that your manuscript has been deemed suitable for publication in PLOS ONE. Congratulations! Your manuscript is now being handed over to our production team.

Kind regards, 

on behalf of

Professor Dr Hoh Boon-Peng 

Academic Editor

PLOS ONE